# Vocal Synchrony of Robots Boosts Positive Affective Empathy

**Shogo Nishimura** [1,*] , **Takuya Nakamura** [1], **Wataru Sato** [2] , **Masayuki Kanbara** [1], **Yuichiro Fujimoto** [1], **Hirokazu Kato** [1] and **Norihiro Hagita** [3]

1    Nara Institute of Science and Technology, Graduate School of Information Science, Ikoma, Nara 630-0192, Japan; nakamura.takuya.nm0@is.naist.jp (T.N.); kanbara@is.naist.jp (M.K.); yfujimoto@is.naist.jp (Y.F.); kato@is.naist.jp (H.K.)
2    Psychological Process Team, Robotics Project, BZP, RIKEN, Kyoto 619-0288, Japan; wataru.sato.ya@riken.jp
3    Art Science Department, Osaka University of Arts, Osaka 585-8555, Japan; hagita@atr.jp
*    Correspondence: nishimura.shogo.nj0@is.naist.jp

**Abstract:** Robots that can talk with humans play increasingly important roles in society. However, current conversation robots remain unskilled at eliciting empathic feelings in humans. To address this problem, we used a robot that speaks in a voice synchronized with human vocal prosody. We conducted an experiment in which human participants held positive conversations with the robot by reading scenarios under conditions with and without vocal synchronization. We assessed seven subjective responses related to affective empathy (e.g., emotional connection) and measured the physiological emotional responses using facial electromyography from the corrugator supercilii and zygomatic major muscles as well as the skin conductance level. The subjective ratings consistently revealed heightened empathic responses to the robot in the synchronization condition compared with that under the de-synchronizing condition. The physiological signals showed that more positive and stronger emotional arousal responses to the robot with synchronization. These findings suggest that robots that are able to vocally synchronize with humans can elicit empathic emotional responses.

**Keywords:** human–robot interaction; affective empathy; nonverbal synchrony; prosodic features

## 1. Introduction

Robots that can converse with humans play increasingly important roles in society [1] by providing automated services, assisting with communication, and offering psychological support. For example, as the number of single-person households increases, feelings of social isolation have increased owing to a decrease in the number of partners available for daily conversation. This can contribute to decreases in happiness and physical activity as well as an increased risk of dementia, particularly among the elderly. To enhance the role of social welfare workers in providing psychological support, dialog robots have recently begun to attract attention. In particular, numerous studies have examined robots that interact with people in daily life [2,3]. However, robots have not yet been designed to interact with people as conversation partners over a long periods. One issue that arises in human–robot conversations is the robot's inability to elicit affective empathy [4,5] from humans.

To address this problem, improvement in human-like countenance in robots should be based on the strategy of imitating human–human communication [6–9]. Here, we focus on vocal synchrony as one feature of such communication, which is defined as the dynamic and reciprocal adaptation of the temporal structure of behaviors between interactive partners [10]. Vocal synchrony is categorized as a nonverbal type of synchrony. Ample evidence in human psychological studies has shown that various types of behavioral synchrony (e.g., gesture mimicking) underlies human empathic interactions [11,12]. Studies have shown that behavioral synchrony heightens empathic feelings including harmonious

relationships [13] and partner likeability [14] as well as empathic behaviors such as help-ing [15]. Several researchers have developed behaviorally synchronized robots and have proposed that such synchrony can produce empathic impressions in human users and might produce empathic understanding in robots [16–22]. Following this concept, several previous studies on robotics have reported that the effects of behavior synchrony (e.g., physical gestures) in robots can facilitate empathic reactions in human users [23,24].

Previous psychological studies have reported that vocal synchrony is associated with empathic understanding in humans [25–27] and that the intentional use of vocal synchrony improves the speaker's affective empathy [25,28,29]. Vocal synchrony in robots is expected to have similar effects [30,31], although no research has verified such results. We hypothe-size that robots with the ability to vocally synchronize with humans during conversation would boost empathic feelings in humans compared with robots not programmed with such an ability.

To test this hypothesis, we developed in this work a robot that can synchronize with the user's voice in terms of vocal prosody, particularly the speech rate, pitch, and volume. To verify the effect of the robot's vocal synchrony, we conducted an experiment in which the robot and human participants conducted predefined conversations with vocal synchrony. The conversations were prepared to induce positive impressions in readers based on preliminary experiments. As the control condition, the robot was de-synchronized from the prosody of participants such that the robot uttered no synchronized voice with comparable prosodic variance. We evaluated the subjective and objective responses regarding affective empathy during the conversation. Specifically, we assessed seven subjective responses related to affective empathy (e.g., emotional connection) using a questionnaire. To complement the subjective rating, which could result in bias such as demand characteristics [32,33], we measured the physiological responses including facial electromyography (EMG) from the corrugator supercilii (i.e., brow lowering) and zygomatic major (i.e., lip corner pulling) muscles as well as the skin conductance level (SCL). The former two and latter measures have been shown to reflect emotional valence and arousal responses, respectively, and psychophysiological evidence suggests that positively arousing events, induce lower, higher, and higher activity in these measures, respectively, relative to emotionally neutral events [34,35]. Based on these rationales and data, we predicted that positive conversations with the robot under the synchronization condition would induce higher subjective affective empathic responses and higher physiological positive and arousing emotional responses compared with that in the de-synchronization condition.

## 2. Related Work

### 2.1. Imitating a Strategy of Human-Human Communication

Several researchers have applied the strategy of human–human communication to interactive agents or robots. Krämer et al. discussed whether a theory specific for human–robot and human–agent interaction is needed and whether theories from human–human interaction can be adapted [6]. Human interaction with agents or robotic systems generally results in actual interaction and relationship building resembling that in human–human communication. They concluded that humans try to apply the forms of interaction and communication in which they are habituated even when agents and robots use other forms of communication. On the basis of such research, there is no need for a theory specific to human–agent/robot interaction. Thus, when designing communication between humans and agents or robots, we should refer to communication factors that humans expect from other humans. Proposing a radically different communication method for agents or robots is difficult for achieving communication with humans. Cassel et al., who developed agents that build a collaborative trusting relationship with users, stated that the various dialog strategies used by humans can also be applied to intelligent agents [7,36]. Edlund et al. discussed many abundant studies on the interaction with a spoken dialog system similar to interaction with a human dialog partner, i.e., as anthropomorphism in spoken dialog systems to attach a human likeness to the systems. They defined "human-like" as "more

like human–human interaction", and they mentioned the need for enhancing the human likeness in spoken dialog system components. In the present study, we develop robots based on the above philosophy.

### 2.2. Synchrony and Affective Empathy

Synchrony is defined as the dynamic and reciprocal adaptation of the temporal structure of behaviors between interactive partners [10]. Bernieri and Rosenthal defined synchrony as "the degree to which the behaviors in an interaction are nonrandom, patterned, or synchronized in both form and timing". It has been argued that the feeling of being "in sync" with a conversational partner underlies many desirable social effects. From a psychological aspect, it is apparent that nonverbal synchrony is associated with empathy in human–human communication [37,38]. Imel et al. revealed evidence for vocal synchrony in a clinical setting in association of synchrony with affective empathy ratings [25]. Ramseyer and Tschacher found that nonverbal synchrony was increased in sessions rated by clients as having high relationship quality and that higher nonverbal synchrony was associated with higher symptom reduction [39]. Such research indicates that the deliberate use of nonverbal synchrony enhances the speaker's assessment of affective empathy.

### 2.3. Affective Empathy between Humans and Robots

Previous robotics studies have reported similar facilitative effects of synchrony on empathic understanding in human–robot interaction by implementing non-vocal synchrony (e.g., physical gestures) in robots [23,24,40]. Specifically, Prepin and Pelachaud [23] proposed a model accounting for the emergence of synchrony depending directly on a shared level of understanding between agents. Their model was based on the human interaction: such that two interactants have similar understanding of the spoken content, their nonverbal behaviors appear to be synchronous. They tested their model through simulation and showed that synchrony effectively emerges between agents having a close level of understanding. They also attributed misunderstanding to de-synchronization. Riek et al. [24] investigated the effect of a user's perceptions during conversation with a robot that mimics the head gesture of the user. They implemented three types of robot that mimicked full head gesture, partial head gesture (only nodding), and only blinking of a user. The comments of participants who interacted with the robot in full and partial mimic conditions indicate that full mimicking can facilitate development a conversational robot that can build rapport with a user. The synchrony of human–robot movements are critical features for evaluating the quality of the imitation. Anzalone et al. [40] analyzed the behavior of robots that perform dynamic joint attention, synchrony of response times, rhythm of interaction and imitation of response times, rhythm of interaction, and variance. They determined that correct comprehension and proper use of nonverbal behaviors are essential for accomplishing optimal interaction to provide readable behaviors and to promote social intelligence in robots.

Furthermore, a few studies have shown that vocal synchrony in robots is expected to have similar effects. For example, Sadouohi et al. reported an approach for creating on-line acoustic synchrony for a robot that can play with a child in a fast-paced, cooperative, language-based game [30,41]. In their study, 40 children played the game with acoustic synchrony applied and not applied to the robot. All children began to enjoy the game over time. However, those who began playing the game with synchronous robots maintained their own synchrony with the robot and achieved higher engagement. Suzuki et al. conducted a psychological experiment to examine the effect of prosodic mimicry by computers on humans [31]. They developed an animated character that mimicked the prosodic features in human voice echoicly by synthesizing hummed sounds. In the proposed method, the character outputs a humming sound in a modulated wave that is combined multiple sine waves to imitate the pitch and volume in human prosody with multiple imitation rates. The results indicate that a higher imitation rate resulted in a better impression of social desirability and familiarity with human characters. However, these

studies did not verify the specific effects based on scientific evidence regarding the effect of voice synchrony by a robot on the affective empathy of humans. In the present study, terms such as "mimicry" and "imitation" are unified as "synchrony".

### 2.4. Research Questions and Hypotheses

We extended the meaning of synchrony and interpret it as vocal synchrony when the spoken dialog system imitates the features of human prosody. Using an approach based on the strategy of human–human communication, we focused on the vocal synchronization and considered its application to dialog robots. We hypothesized the effect of vocal synchrony of robots that can enhance the positive affective empathy in humans, and we tested the following hypotheses by comparing robots that speak vocally in synchronization with humans in conversation and robots not programmed with such an ability.

**Hypothesis 1.** *The vocal synchrony of robots would enhance users to empathic responses subjectively.*

**Hypothesis 2.** *the vocal synchrony of robots would boost physiologically arouse more positive and stronger emotions in the user's response.*

To test these hypotheses, we used a robot that observes the features of human voice in real-time and speaks on the basis of the aforementioned prosodic features.

### 3. Implementation of the Vocal Synchrony

We developed a computer system that can synchronize with humans in terms of vocal prosody. In previous research, features such as speech rate [42], pitch [43], volume [43], accent [44,45], utterance length [46], response latency [47], pausing frequency and length [46,47], and laughter [46] have been discussed. For the present study, we selected three features: speech rate, pitch, and volume. We developed a system that can estimate these three features in a human voice to generate a robot voice synchronizing these features. This system was executed dynamically, and the robot voice was applied to the robot utterance immediately after a person spoke.

### 3.1. Details of the Estimation Methods for Prosodic Features

As voice analysis and speech synthesis tools, our system uses Julius [48], which is an open-source large vocabulary continuous speech recognition engine and Speech Signal Processing Toolkit (SPTK) [49] as a voice analysis tool. In addition, VoiceText [50] was used as a speech synthesis tool. The process from the recognition of a human utterance to the generation of the robot voice is as follows. (1) The user's speech is recognized by Julius, (2) the speech rate is estimated by Julius, (3) the pitch and volume are estimated by SPTK and are used as the prosodic features, and (4) each estimated value is converted to a voice synthesis parameter specific to VoiceText, which performs the voice synthesis.

The Japanese equivalent of "beat" is "mora", and each mora is temporally equivalent. In Julius, which estimates the speech rate, the average number of mora per second obtained by dividing the number of mora by the length of the utterance interval obtained in the preprocessing of speech recognition processing was used as a feature value representing the speech rate. Essentially, it has the structure of a consonant and vowel, although exceptions exist such as "n", which has no vowels, and "tsu", which is applied to the sound itself [51]. In SPTK, which estimates pitch and volume, the fundamental frequency ($F0$) of the wave file obtained from the human utterance is extracted, and the average pitch is estimated by calculating the average value in the logarithmic domain. Then, the analysis window is shifted every 10 μs, and $F0$ is output. The average value of $F0'$, obtained by removing the unvoiced sound section from the $F0$ value obtained here, is estimated as the human pitch. To estimate the volume, the average value of the power term in the speech segment is used for simplicity, which is estimated as the volume of the human. In VoiceText, however, speech synthesis is performed using both male and female speech models. The voice outputted by speech synthesis for each gender is calibrated on the basis of the estimated

prosodic features. The calibration formula, determined from the estimated human prosodic features of the VoiceText speech synthesis parameters, is described below.

VoiceText has limited parameters of 50–400% speech speed and 50–200% pitch and volume. Text data are required to generate synthesized speech, such as the following five specific Japanese sentences [52].

- あらゆる現実をすべて自分のほうへねじまげたのだ / I twisted all the reality toward myself. (Arayuru genjitsu wo subete jibun no hou e nejimageta noda)
- テレビゲームやパソコンでゲームをして遊ぶ / I play video games and games on my computer. (Terebi ge-mu ya pasokon de ge-mu wo shite asobu)
- 救急車が十分に動けず救助作業が遅れている / Ambulance is not moving enough, and rescue work is delayed. (Kyuukyuusha ga jyubun ni ugokezu kyuujyosagyou ga okureteiru)
- 老人ホームの場合は健康器具やひざ掛けだ / In case of a nursing home, it is a health appliance or a rug. (Roujin ho-mu no baai wa kenkoukigu ya hizakake da)
- 嬉しいはずがゆっくり寝てもいられない / I should be happy, but I can't sleep slowly. (Ureshii hazuga yukkuri netemo irarenai)

These five sentences contain various phonemes of well-balanced distribution and are considered suitable for our system in supporting various utterances. A calibration formula was derived for each speech synthesis parameter of speech rate, pitch, and volume. The derivation procedure is shown below.

(i) The corresponding voice synthesis parameter of VoiceText is set to 50 (speech speed is 70) and outputs synthesized speech. At this time, the parameters other than those used for deriving the calibration formula are fixed to 100.
(ii) The synthesized speech output from the Nao humanoid robot's speaker is input using a microphone, and the prosodic features are then estimated. At this time, the volume setting of Nao is fixed at 65%.
(iii) The estimated prosodic features and the speech synthesis parameters are recorded.
(iv) Steps (i–iii) are repeated three times for each of the five sentences.
(v) The corresponding speech synthesis parameters are changed by 10, and steps (i–iv) are conducted.
(vi) Steps (i–v) are repeated until the parameter is 200 and the speech rate is 150.

The calibration formula was derived on the basis of the relationship between the speech synthesis parameters and the prosodic features obtained from the above process. As speech models provided by VoiceText, we used "takeru" for male models and "hikari" for female models. The accuracies of our proposed estimation methods are shown in Appendix B.

### 3.2. System Flow of Vocal Synchrony

Figure 1 shows the flow beginning when the system receives a user's voice to our model output of the synchronized voice. After receiving the voice input from a user through a microphone, the system performs three major processes successively. First, the system estimates the prosodic features by Julius and SPTK in which the three features of speech rate, pitch, and volume are quantified. Second, based on the quantified values, each parameter for generating the synthesized voice is converted by our model, as shown in Table 1. At this time, it is assumed that the gender of the user is known, and it is necessary to manually set in advance the gender of the synthesized voice to be generated. Third, a synthesized voice is created by VoiceText, and the voice file is sent to the robot. The above process takes about 2–3 s and is designed to be applicable to any type of network robot.

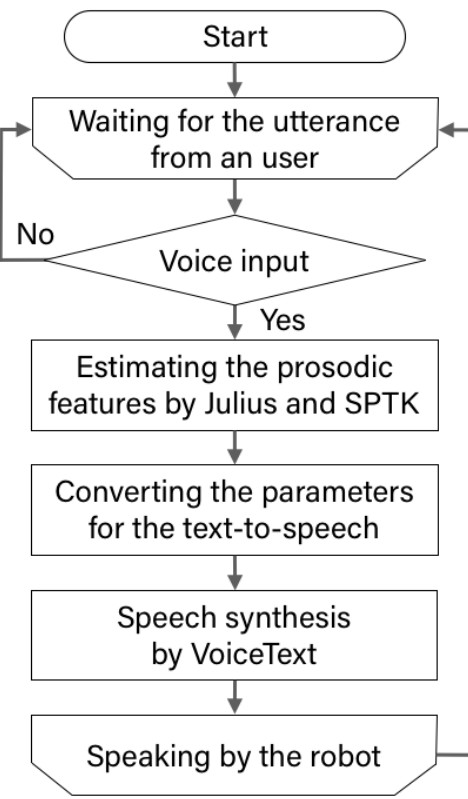

**Figure 1.** The processing flow of our model.

**Table 1.** Calibration formulas for each prosodic feature and gender.

| | Voice Model | |
|---|---|---|
| | **Takeru: Male Model ($R^2$)** | **Hikari: Female Model ($R^2$)** |
| Speech rate | $y = 10.84x + 8.73$ (0.91) | |
| Pitch | $y = 1.01x - 0.18$ (0.99) | $y = 1.01x - 0.48$ (0.99) |
| Volume | $y = 0.25x - 2.98$ (0.98) | $y = 0.26x - 3.23$ (0.95) |

## 4. Experimental Conditions

### 4.1. Participants

We tested 27 Japanese volunteers including 15 females and 12 males with a mean age $\pm$ standard deviation (*SD*) of $23.0 \pm 5.7$ years. The required sample size was determined using a priori power analysis based on G*Power software version 3.1.9.2 [53]. We conducted Student's univariate *t*-tests (one-tailed) for physiological responses with $\alpha$ level of 0.05 and power of 0.80. The effect size ($d = 0.5$) was estimated on the basis of a preliminary experiment using a different sample ($n = 5$) with a similar procedure and facial EMG recording. The result of the power analysis showed that 27 participants were needed. The participants were recruited through advertisement at Kyoto University, and each participant received 1000 Japanese yen book coupons. After a detailed explanation of the experimental procedure, all participants provided informed consent. The study was approved by the local ethics committee of Nara Institute of Science and Technology. The experiment was conducted in accordance with institutional ethical provisions and the Declaration of Helsinki.

### 4.2. Experimental Design

A one-factor within-subjects design was employed, with the vocal synchrony condition (synchronization versus de-synchronization) as a factor. Dependent variables included six subjective ratings: feeling friendliness, feeling fun, feeling connection, feeling synchrony,

motivation to interact, and feeling active listening; one subjective preference selection; and three physiological responses: corrugator supercilii EMG, zygomatic major EMG, and SCL.

### 4.3. Apparatus

An electric condenser microphone (ECM-674/9X, Sony Corporation, Tokyo, Japan) was used as the system audio input device. For the output, a dialog robot was used with the built-in speaker of the Nao robot v. 4 [54] in the impression evaluation. This selection was made because its appearance does not have gender information; therefore, it can naturally synchronize with both male and female participants.

We used sets of pre-gelled, self-adhesive 0.25 cm Ag/AgCl electrodes (Prokidai, Sagara, Japan) and an EMG-025 amplifier (Harada Electronic Industry, Sapporo, Japan) for the EMG recording; pre-gelled, self-adhesive 1.0 cm Ag/AgCl electrodes (Vitrode F, Nihonkoden, Tokyo, Japan) and a BioDerm Skin Conductance Meter (Model 2701 UFI, Morro Bay, California, United States) for SCL recording; and a digital web camera (HD1080P, Logicool, Tokyo, Japan) for unobtrusive video monitoring. For sampling the data, we used the PowerLab 16/35 data acquisition system and LabChart Pro v. 8.0 software (AD Instruments, Dunedin, New Zealand) for sampling the data.

### 4.4. Dialog Scenario

We created 12 scenarios for conversations with the robot. Predefined scenarios were chosen rather than chats to control the verbal and emotional content in the conversation. Each scenario was structured such that the robot and the participant each took seven turns alternately and that the person reading the scenario had a positive impression. An example is shown in Figure 2. Some of other samples are shown in Figures A1 and A2.

To prepare the scenarios for producing positive impressions in readers, we preliminary created 14 scenarios. Then, we conducted a preliminary rating experiment with seven male Japanese participants, none of whom took part in the subsequent robot experiment. The participants read each scenario and evaluated it using the affect grid [55]. On the basis of the results, we selected 12 scenarios having positive valence and high arousal.

These 12 scenarios were randomly divided into 3 sets of 4 scenarios, and each participant evaluated a randomly selected set. The topics in each scenario set are described in Appendix A.

| | |
|---|---|
| **Nao** | **: Hello!** |
| **Participant** | **: Hello!** |
| **Nao** | **: What is your favorite food?** |
| **Participant** | **: I really like ramen.** |
| **Nao** | **: I love it too! What type of ramen do you like?** |
| **Participant** | **: I like soy sauce ramen.** |
| **Nao** | **: I often go to eat soy sauce ramen!** |
| **Participant** | **: Shall we go to eat it together?** |
| **Nao** | **: Really? I do want to go with someone!** |
| **Participant** | **: Actually, I know a good ramen restaurant.** |
| | **  (like a secret talk)** |
| **Nao** | **: Where is it? Please let me know.** |
| **Participant** | **: It is in Osaka. I will show it to you next time!** |
| **Nao** | **: Thank you! I am looking forward to going there!** |
| **Participant** | **: Yes, me too!** |

**Figure 2.** Sample of text showing colored lines. The participants were instructed to speak with positive and stronger emotion (negative emotion and sleepiness) when reading the red (blue) text.

### 4.5. Procedure

The experiments were conducted individually with each participant. Upon arrival, the outline of the experiment was explained to each participant, who was asked to sign a consent form. Before interacting with the robot, the participants were told that electrodes would be placed on their bodies to record sweat gland activity, which concealed the actual purpose for facial EMG recording. Electrodes for facial EMG from the corrugator supercilii and zygomatic major muscles were placed on the left part of the participant's face; for SCL, they were placed on the palmar surface of the medial phalanges of the index and middle fingers of the left hand. The participants were instructed on how to interact with the robot by reading the scenarios. Next, the experiment was prepared for practice. During the experiment, the participants were handed a written practice scenario and were instructed to read the text aloud with emotion, according to the instructions on the page. The sentences in the scenario were arbitrarily colored red to indicate the points at which the participants should speak with more positive and stronger arousal emotion and blue to indicate when they should speak with negative and sleepiness emotion, as shown in Figure 2. We asked the participants to watch a video showing an example of the interaction with the robot that was prepared in advance to more accurately convey the instructions. Then, the participants entered a soundproof room (Science Cabin, Takahashi Kensetsu, Tokyo, Japan) and faced the robot to begin the experiment, as shown in Figure 3.

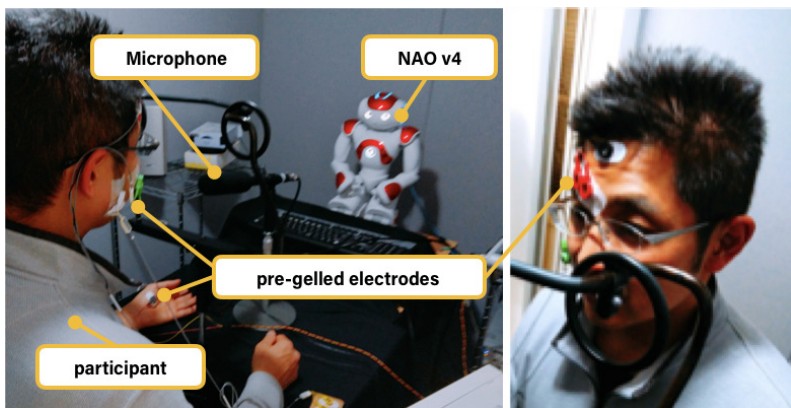

**Figure 3.** Photograph of the experimental setup. Participants read dialog scenarios in the presence of the Nao robot v. 4. The participant's voice was recorded using a microphone, and the facial electromyography and skin conductance level were recorded using electrodes. The subjective responses of the participants were also assessed using questionnaires.

The participants practiced reading two passages that were not used in the experiment; the robot was not present during this exercise. Finally, the participants were requested to read the practice scenario again in the presence of the robot, and their voice data were acquired to prepare the robot's voice under the de-synchronization condition. For this purpose, the average values of the three prosodic features were calculated for each participant. In the de-synchronized condition, our system generated the prosodic feature of robot's output based on these average values. The system calculated the distance between the values from the voice of participant for each prosodic feature and the average values. The distance was applied by adding that value to the average value in the opposite direction as the output of the robot.

During the participant's dialog with the robot and questionnaire session, the experimenter was not in the same room. Each participant read the four scenario sets twice in the presence of the robot with and without the robot's vocal synchronization. A example of dialog between a participant and the robot is shown in Figure 4. The scenario order and synchronization conditions were randomized among the participants.

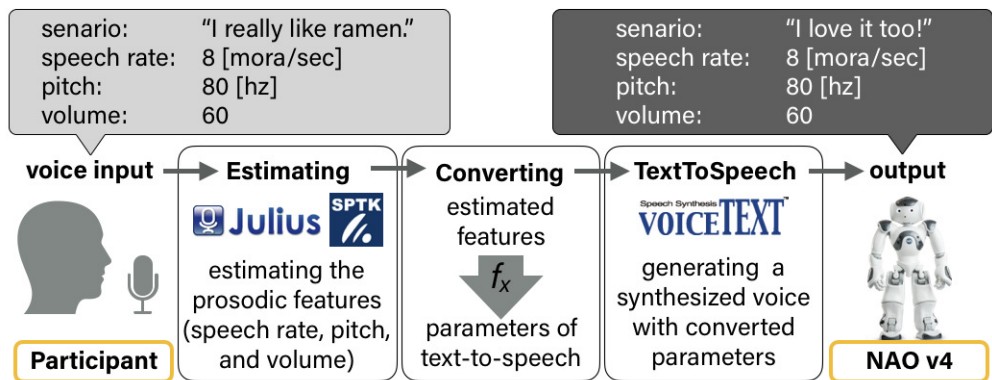

**Figure 4.** Example of dialog between a participant and the Nao robot in the experiment and system operation flow under the synchronization condition.

### 4.6. Questionnaire

The questionnaire used to measure participant's subjective responses was designed to measure affective empathy. After each reading, the participant completed the questionnaire for evaluating the robot by rating the following six features on the seven-point Likert scale, in which a score of 1 means "not at all", and that of 7 means "very much": friendliness, enjoyability, emotional connection, degree of synchronization, motivation to interact with the robot, and degree of active listening. Participants were also asked to comment qualitatively on their feelings about interacting with the robot; this request was not mandatory. The answers were handwritten in a private room.

After completing each scenario, the participants were asked to indicate their favorite condition. The questions were prepared following those of previous studies designed to evaluate constructs related to affective empathy [56–58]. The questionnaire included the following statements.

1. Did you feel friendliness?
2. Did you feel fun?
3. Did you feel that the robot was listening to you?
4. Did you feel an emotional connection?
5. Did you feel the motivation to use?
6. Did you feel synchrony?
7. Which robot did you like (after finishing each scenario)?

During the experiment, the physiological signals from the participants were continuously recorded. The EMG data were recorded from the corrugator supercilii and zygomatic major muscles through electrodes with 1 cm interelectrode spacing. A ground electrode was placed on the forehead. The data were amplified and filtered online at a bandpass of 20–400 Hz [59]. The SCL was measured by applying a constant voltage of 0.5. The data were recorded with no online filter and were sampled at 1000 Hz.

### 4.7. Data Analysis

4.7.1. Preprocessing

For the subjective responses, the mean values for each scale were calculated and analyzed using Student's one-tailed *t*-test to contrast the synchronization and de-synchronization conditions.

For the physiological data, preprocessing was performed using Psychophysiological Analysis Software 3.3 (Computational Neuroscience Laboratory of the Salk Institute; La Jolla, CA, USA) and in-house programs implemented in MATLAB 2018 (MathWorks, Natick, MA, USA). The data were sampled for 1 s during the pre-conversation baseline and during the conversation in each trial. One of the authors blindly checked the video, raw EMG, and raw SCL data and rejected the artifact-contaminated trials. To evaluate the artifacts in the video data, we used the artifact lists given in a previous study [60], e.g.,

head swinging and guidelines [61]. The percentages of artifact-contaminated trials showed no significant systematic differences between the two conditions, with the mean $\pm SD =$ 4.63 $\pm$ 6.67 and 6.17 $\pm$ 7.52 for the synchronization and de-synchronization conditions, respectively; $t$-test, $p > 0.1$. For each trial, the EMG data were rectified, baseline corrected, and averaged across the conversation period. The SCL data were processed in the same manner except they were not rectified.

### 4.7.2. Statistical Analysis

Statistical analysis was performed using SPSS 16.0J (SPSS Japan, Tokyo, Japan). We performed multivariate analysis using the paired-sample Hotelling's $T^2$ test, which is a multivariate generalization of the univariate paired-sample $t$-test, to differentiate between the synchronization and de-synchronization conditions using all dependent variables to control the experiment-wise type I error rate [62]. For the follow-up univariate analyses, we conducted paired-sample $t$-tests (one-tailed) and tested our predefined expectations on the differences between these conditions. We conducted preliminary analyses using the factor of sex and found no significant main or interactive effects in the subjective and physiological responses (two-way analysis of variance, $p > 0.1$). Therefore, this factor was disregarded.

## 5. Results

First, we tested the effects of synchronization versus de-synchronization on all responses using Hotelling's $T^2$ test and found a significant effect of $T^2$ (10, 17) = 5.30, $p < 0.005$. Then, we evaluated these effects on each subjective and physiological response using univariate $t$-tests.

### 5.1. Subjective Responses

All of the subjective responses to the questionnaire were significantly higher in the synchronization condition than in the de-synchronization condition ($t$-test, $p < 0.001$; Figure 5 and Table 2). In the item of feeling friendliness derived from the question "Did you feel friendliness?", there was significant difference in the mean value between the synchronization condition and the de-synchronization condition ($t$-test, $p < 0.001$). In the items regarding feeling friendliness, feeling fun, feeling an emotional connection, feeling synchrony, feeling motivation to interact, feeling of being actively listening to, and robot preference, significant differences were noted in the mean value between the synchronization condition and the de-synchronization condition ($t$-test, $p < 0.001$).

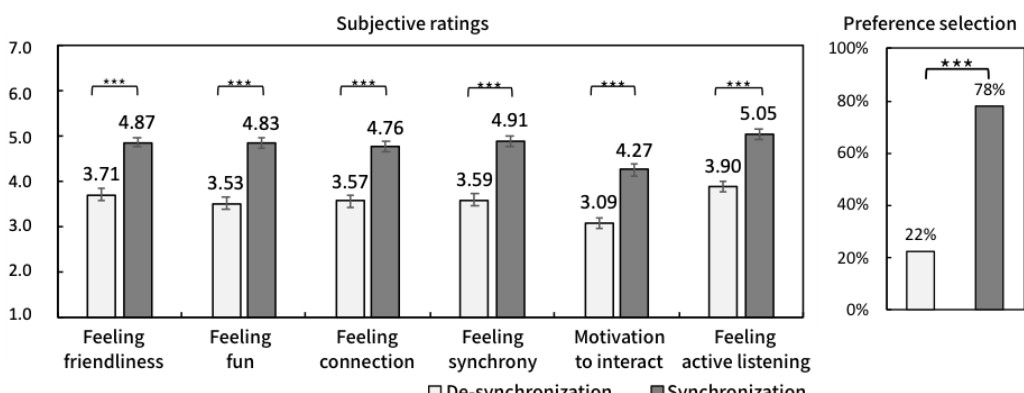

**Figure 5.** Mean subjective ratings with standard error (**left image**) and preference selection (**right image**); *** $p < 0.001$.

**Table 2.** Results of one-sample *t*-tests for the questionnaire responses.

| Measure | Statistic | | |
|---|---|---|---|
| | *t* | *p* | *d* |
| Feeling friendliness | 7.02 | $1.1 \times 10^{-10}$ | 1.36 |
| Feeling fun | 8.43 | $8.6 \times 10^{-14}$ | 1.38 |
| Feeling connection | 7.35 | $2.1 \times 10^{-11}$ | 1.30 |
| Feeling synchrony | 7.40 | $1.7 \times 10^{-11}$ | 1.43 |
| Motivation to interact | 7.53 | $8.4 \times 10^{-12}$ | 1.24 |
| Feeling active listening | 6.99 | $1.2 \times 10^{-10}$ | 1.28 |
| Preference selection | 6.91 | $1.8 \times 10^{-10}$ | 1.88 |

### 5.2. Physiological Responses

The physiological responses (Figure 6) of the corrugator supercilii EMG, zygomatic major EMG, and SCL differed significantly. In particular, they were lower, higher, and higher under the synchronization conditions, respectively, than those measured under the de-synchronization condition (*t*-test, $p < 0.05$; Table 3).

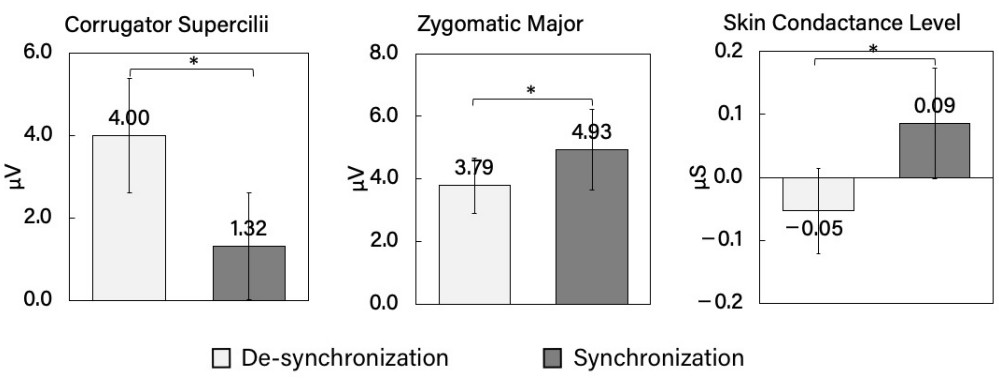

**Figure 6.** Means with standard error of the facial electromyography from the corrugator supercilii (left image), zygomatic major muscles (middle image), and skin conductance level (right image); * $p < 0.05$.

**Table 3.** Results of one-sample *t*-tests for the intra-individual correlation coefficients of the expected subjective physiological concordance across the stimuli.

| Measure | Statistic | | |
|---|---|---|---|
| | *t* | *p* | *d* |
| Corrugator supercilii EMG | 2.02 | 0.027 | 0.54 |
| Zygomatic major EMG | 1.73 | 0.047 | 0.28 |
| SCL | 1.72 | 0.049 | 0.48 |

EMG: electromyography; SCL: skin conductance level.

## 6. Discussion

The results of the subjective responses consistently indicated that the participants felt stronger affective empathy-related feelings in response to the robot that vocally synchronized with them compared with the robot that did not. These results supported out H1. Moreover, those results are consistent with previous psychological findings such that vocal synchronization enhances empathic communications in humans [25,29], and they agree with other findings from previous robotics studies [23,24]. However, because no previous study has investigated the effects of a robot's vocal synchronization on conversations with humans, to the best of our knowledge, the present study provides the first evidence of robot synchronization with a human voice to boost empathic feelings.

The participants' physiological responses showed that the synchronization condition induced lower corrugator activity, higher zygomatic activity, and higher SCL. Based on the ample evidence regarding the association between these physiological measures and subjective emotional responses [34,35], the facial EMG data jointly indicate more positive valence, and the SCL data indicate stronger arousal in response to the robot that vocally synchronized with them compared with the robot that did not. These data complement the subjective ratings and provide objective evidence that a robot that can vocally synchronize with users can enhance empathic responses. These results supported out H2.

These findings have practical significance for the development of dialog robots. Specifically, it might be possible to facilitate human–robot rapport by equipping robots with vocal synchrony ability. This can enable robots to gain social value as conversation partners because it will be easier for users to accept active assertions from robots. In turn, users can realize other benefits such as behavior modification or self-reliance support.

These results also have theoretical implications for human empathic communication. Although several previous psychological studies of humans have suggested that vocal synchrony can induce empathic feelings [28], other research failed to indicate such an effect [25,29]. This discrepancy can be attributed at least partially to the difficulty in implementing vocal synchrony in humans while controlling other confounding variables (e.g., volume). Our experiment with the robot clearly shows that vocal synchrony boosts empathic feelings during conversation.

Among the participants who favored the de-synchronization condition, no negative opinions toward the scenarios were indicated; however, some impressions toward the voice of the robot were observed. One participant provided the following free descriptions regarding the robot under the synchronization condition: "The conversation was fast, and I did not get a good impression"; "the robot talked to me according to my way of speaking, but it felt like I was concerned, and it was difficult to talk"; and "one of the topics was good, but in the other topic, I felt that the robot seemed boring." On the basis of these opinions, we determined that in some cases, the de-synchronization condition might lead to better impressions with a participant than that occurring under the synchronization condition. It was shown that any strangeness or discomfort experienced by a participant during the interaction worsened the impression of the robot. Therefore, a more accurate tuning vocal synchrony technique is required for implementation. However, according to the results of this experiment, our proposed method for measuring human voice and generating a synthesized voice in real-time was sufficiently effective, and the validity of the technology was demonstrated.

To address one of the implementation issues, we need to consider the control of the de-synchronization condition in the experiment. The proposed system calculates the distance between the values from the voice of the participant for each prosodic feature and the average values. In the de-synchronization condition, it applies the distance by adding that value to the average value in the opposite direction for each prosodic feature as the output of the robot. In the experiment, we instructed the participants to add inflections to their speech as a control to make it easier for the participants to feel the changes in the prosody of the robot. According to the opinions of each participant and the observation of the video during the experiment, the prosody did not reflect the rise or fall of emotion in the participant's speech in all cases.

The following comments were obtained from the participants regarding the robot de-synchronization condition: "There was an inflection in the way the robot talked, and it was interesting", "I felt a lot of intonation in terms of volume", and "I felt there was a bigger difference between excited or depressed feelings in the robots than former one." Thus, under the de-synchronization condition, the participant perceived the intonation of the robot. To resolve this problem, the prosodic information of the robot can be fixed to a certain value under the de-synchronizing condition.

Several limitations of this study should be acknowledged. First, we used only de-synchronization as the control condition, which produced no synchronization with compa-

rable prosodic variance in the robot's voice. Further experiments including different control conditions are needed to confirm the findings. Specifically, because the de-synchronization lacked positive emotional prosody, whereas the synchronized condition included this owing to synchronization of the participant's prosody, a control condition without synchronization but with positive prosody is needed. Second, we investigated only the positive emotional interaction, upon which we made predictions based on previous evidence; further studies are needed to investigate the negative emotions. This issue is particularly important because robots can be used in psychotherapy if they can downregulate human negative emotions [63]. Interestingly, the data of human vocal synchrony in psychotherapy are mixed. One study reported a positive association between vocal synchrony and the empathy rating in a therapy setting [25], whereas a different study reported a negative association [64]. It was proposed that reduction of vocal synchrony can dampen a patient's negative emotions under some situations [65]. Future studies would benefit from investigating whether the robot's vocal synchrony can up- or downregulate human negative emotions. Third, we used the scenarios to evaluate the robot in controlling the reproducibility in the quality of the dialog. However, this dialog was different from daily conversation. As a future task, the robot should be evaluated based on free conversation.

## 7. Conclusions

We validated whether a robot can boost the positive affective empathy of a user according to subjective and physiological empathic responses in scenario-based interactions. Evaluation experiments were conducted using a robot programmed with a system for analyzing the prosodic features of user's voice and the ability to generate a synthesized voice including the estimated features. The subjective ratings consistently revealed heightened empathic responses to the robot under the synchronized condition compared with that under the de-synchroniziation. Moreover, the physiological signals indicated more positive and stronger arousal emotional responses to the robot with synchronizing. These data suggest that robots with the ability to vocally synchronize with humans can elicit empathic emotional responses in the humans.

**Author Contributions:** Conceptualization, S.N., W.S., and M.K.; data curation, S.N., T.N., and W.S.; formal analysis, S.N., T.N., and W.S.; writing—original draft, S.N. and W.S.; writing—review and editing, M.K., Y.F., H.K., and N.H. All authors have read and agreed to the published version of the manuscript.

**Funding:** A part of this research was supported by the JSPS Foundation 18H03274 and Research Complex Promotion Program.

**Institutional Review Board Statement:** The study was conducted according to the guidelines of the Declaration of Helsinki, and approved by the Institutional Review Board (or Ethics Committee) of Nara Institute of Science and Technology (2019-I-8, approved on 29 July 2019).

**Informed Consent Statement:** Informed consent was obtained from all participants involved in the study.

**Conflicts of Interest:** The authors declare no conflict of interest.

## Appendix A. Content of Scenarios

Three topics for each scenario set used to evaluate the robot were given for each scenario set. The three topics in each scenario set are described below.

**Scenario set 1:** self-introduction, favorite food, and movie invitation

**Scenario set 2:** self-catering, dinner invitation, and lack of sleep

**Scenario set 3:** favorite sports, summer vacation events, and sauna

**Scenario set 4:** part-time job, long vacation, and lost wallet

A sample from the entire content regarding the topic of self-introduction given in scenario set 1 is shown in Figure 2. Other samples of the topic contents are shown in Figures A1 and A2.

Nao : Hello!
Participant : Hello!
Nao : What are you doing on your days off these days?
Participant : I have no plans lately, so I'm free.
Nao : Then why don't you go see a movie next time?
Participant : It is good! Let's go!
Nao : How about this Sunday?
Participant : Well, there are plans for this Sunday.
Nao : Oh, is that so?
Participant : How about this Saturday?
Nao : I can go! Well then, let's do it next Saturday.
Participant : Yes! Let's have lunch and go for the movie.
Nao : All right! Then, let's meet in front of the station at 12 o'clock.
Participant : I got it.

**Figure A1.** Content regarding the topic of movie invitation in scenario set 1.

Nao : Hello!
Participant : Hello.
Nao : You're getting tired lately.
Participant : I'm so busy with work that I can't get rid of my fatigue.
Nao : It's hard. Then how about a Sauna?
Participant : I don't like it. What's good?
Nao : If you sweat a lot, your fatigue will disappear!
Participant : It's so hot that I can't stand it.
Nao : That's right. Can I tell you my favorite method enjoying sauna?
Participant : Is there such a thing? Please let me know!
Nao : Yes! If you know the correct way to enjoy sauna, you will love it!
Participant : If you don't mind, would you please go with me next time?
Nao : Let's definitely go! I'll introduce my recommended public bath!
Participant : Thank you so much!

**Figure A2.** Content regarding the topic of sauna in scenario set 3.

## Appendix B. Accuracy of Vocal Synchrony

We evaluated the accuracy of our voice synchronization system. Specifically, the similarity between human voice and synthetized voice generated using our model is evaluated subjectively and objectively. The evaluation method included the following steps.

1. An experimenter recorded the voices of seven humans with various prosodic features (i.e., fast, slow, high, low, large, small, neutral). The content of the utterance was "あらゆる現実をすべて自分のほうへねじまげたのだ / I twisted all the reality toward myself."
2. Each voice was estimated using the prosodic parameters (i.e., speech rate, pitch, and volume).
3. Trials 1 and 2 were repeated five times each, and the average value for each voice was used as the prosodic feature of a human in each voice.
4. Based on the prosodic features of a human, seven types of synthesized voices of robots were generated using VoiceText.
5. Similar to that for the human voice, the average of five trials was used as the prosodic feature of the robot.

We recruited participants through advertisement at Nara Institute of Science and Technology, and each participant received 1000 Japanese yen. We asked eight male participants

to evaluate the parameters of speech rate, pitch, volume, and voice as a whole using the seven-point Likert scale (1: Not similar at all, 7: Completely similar.): The participants were requested to determine the similarity between the voices of seven groups of humans and robots. This experiment was performed before main experiment described in Section 4. The results of the subjective evaluation are shown in Figure A3.

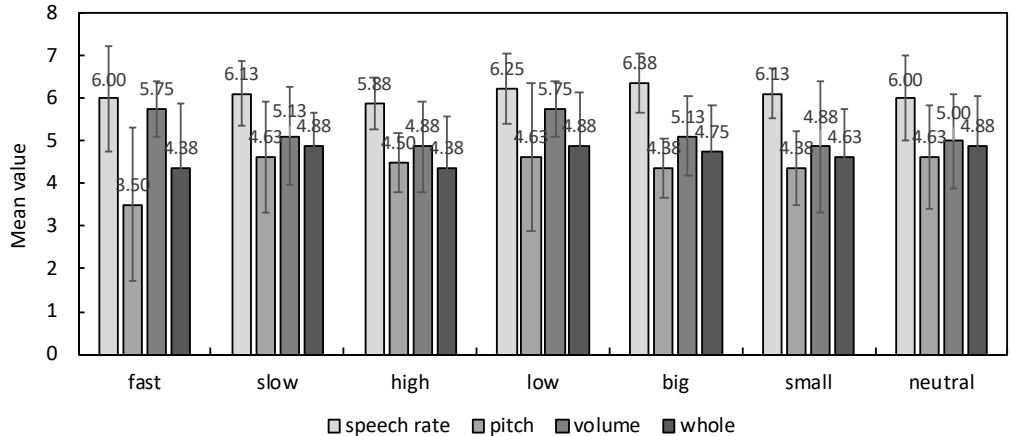

**Figure A3.** Mean value of subjective responses.

As shown in the Figure A3, the voice should be the main focus. In particular, it sufficiency met the condition for our model application because the mean values were above 4 for all the samples. Next, we calculated the cosine similarity and the correlation coefficient between the voices of seven groups of humans and robots to evaluate the objective aspect.

As a result, all of the samples showed sufficiently high cosine similarity and correlation coefficients through the objective evaluation, as shown in Table A1. In this experiment, we recorded the voices from male experimenters; thus, the results showed the similarity of synthesized voice using only a male to model of the human voice. Based on this result, we expect that the similarity for the female model will be sufficiently evaluated. However, our sample was small, and we did not perform tests for both genders. Future studies using a larger sample size might be needed to confirm the findings.

**Table A1.** Mean values of prosodic features, cosine similarity, and correlation coefficient.

| Sample | Speaker | Speech Rate | Pitch | Volume | Cosine Similarity | Correlation Coefficient |
|---|---|---|---|---|---|---|
| Fast | HUMAN | 124.24 | 98.16 | 101.02 | 0.9999 | 0.9954 |
| | ROBOT | 125.00 | 98.05 | 103.59 | | |
| slow | HUMAN | 83.43 | 98.74 | 100.53 | 0.9997 | 0.9944 |
| | ROBOT | 81.11 | 98.93 | 103.68 | | |
| high | HUMAN | 98.53 | 123.85 | 102.76 | 0.9998 | 0.9799 |
| | ROBOT | 97.30 | 123.96 | 106.77 | | |
| low | HUMAN | 102.03 | 79.13 | 98.75 | 0.9999 | 0.9969 |
| | ROBOT | 98.53 | 78.95 | 97.33 | | |
| large | HUMAN | 101.58 | 98.78 | 124.22 | 0.9994 | 0.9905 |
| | ROBOT | 97.71 | 99.06 | 129.51 | | |
| small | HUMAN | 101.58 | 98.81 | 73.82 | 0.9998 | 0.9936 |
| | ROBOT | 98.53 | 99.14 | 75.46 | | |
| neutral | HUMAN | 97.71 | 99.33 | 99.97 | 0.9999 | 0.8868 |
| | ROBOT | 97.71 | 99.03 | 102.18 | | |

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
