# Peer review of "Vocal Synchrony of Robots Boosts Positive Affective Empathy"

_applsci, doi:10.3390/app11062502_

Round 1

Reviewer 1 Report

This paper studied whether vocal synchronization in a robot affects emotional responses of those who interact with it. An experiment is presented with two conditions: vocal synchronization and de-synchronization. The robot was evaluated through a questionnaire, as well as using physiological signals gathered from the participants. The paper concludes that vocal synchronization can increase emphatic emotional responses.

On the plus side, the topic is important and some interesting and novel ideas are presented. Further, statistical analyses are properly discussed. However, I do not believe that the paper can be accepted at this current state: (a) additional experiments are needed to further study the proposed research question and to investigate possible confounding factors, and (b) more information about the methodology and results is needed to be able to verify the validity of some of the conclusions. My comments are below.

The topic needs to be better motivated in the introduction and some claims in the introduction need to be backed up by the literature. Currently, the introduction contains some details about the methodology that do not need to be included in the introduction. Rather, it needs to motivate the research questions further based on the existing literature.

Research questions and hypotheses are not specified. A section should be added specifying research questions and hypotheses.

Background should be expanded to include more literature, for example on robots’ non-verbal synchronization, implementation of emotions in robots’ speech, and impact of social behaviours of robots on humans.

English translation should be provided for the sentences on page 4.

Was the synchronization method proposed in the paper tested? It should have been tested separately to ensure that the synchronization condition was properly implemented.

The methodology needs to be explained in more details. It is not possible to evaluate the methodology given the current information. For example, the paper provides qualitative results in the discussion (participants’ comments), while it is not clear when and how the participants were asked for these comments. The de-synchronization condition needs to be further explained. The procedure also needs to be further explained. What were the participants asked to do when entering the soundproof room? How was the order randomized in each scenario (between or within participants?). Did the participants read all the sentences with emotions? Who did they practice the scenarios with in the first step?

Design of the questionnaire should be also further explained.

While the paper studied the effect of some factors such as gender, order effect, which is important in this study was not controlled for.

As the authors discuss, lack of a control condition, as well as lack of negative scenarios are important limitations of this study. I believe that without another study that involves a control condition and negative scenarios, it is not possible to draw conclusions about the effect of vocal synchronization. Therefore, a follow up study can help improve this paper. Also, some of the conclusions are very strong. For example, the Conclusion section starts with: “We developed a robot that can synchronize with human voice”. However, this is not accurate, as neither a robot was developed nor the ability of synchronizing with humans was independently evaluated.

Another limitation that is not discussed in the paper but is important, is that the scenarios were given to the participants and read by them. This is different in nature from actual interactions. For example, there would be more mental workload as a result of reading the scenarios and processing the emotional instructions, which can also affect participants’ attention to the robot.

There are multiple errors that need to be fixed (e.g., “to improve robot’s human-like…” -> robots’; exited -> “excited”)

Reviewer 2 Report

The authors show an experiment where the robot can synchronize with the human voice.

Two conditions are tested with and without vocal synchronization in order to show the impact of synchronization on interaction. The results show that the robot can vocally synchronize with humans and elicit empathic emotional responses.

I have several comments:

(1) Overall, the paper lacks clarity. You must absolutely explain the following points in more detail:

  • Details of the estimation methods for prosodic features 
  • Dialog Scenario
  • Data Analysis
  • Figure 2

(2) I recommend to add some references to improve the bibliography:

  • Asada, M. (2015). Towards artificial empathy. International Journal of Social Robotics7(1), 19-33.
  • Revel, A., & Andry, P. (2009). Emergence of structured interactions: From a theoretical model to pragmatic robotics. Neural networks22(2), 116-125.
  • Boucenna, S., Gaussier, P., Andry, P., & Hafemeister, L. (2014). A robot learns the facial expressions recognition and face/non-face discrimination through an imitation game. International Journal of Social Robotics6(4), 633-652.
  • Andry, P., Blanchard, A., & Gaussier, P. (2010). Using the rhythm of nonverbal human–robot interaction as a signal for learning. IEEE Transactions on Autonomous Mental Development3(1), 30-42.
  • Delaherche, E., Boucenna, S., Karp, K., Michelet, S., Achard, C., & Chetouani, M. (2012, November). Social coordination assessment: Distinguishing between shape and timing. In IAPR Workshop on Multimodal Pattern Recognition of Social Signals in Human-Computer Interaction (pp. 9-18). Springer, Berlin, Heidelberg.

(3) In you paper, you use synchrony, mimicry and imitation. It will be interesting to define and discuss these terms. 

(4) In your paper, the model of synchronisation is unclear. I recommend having a section and a figure that explains it.

(5) The estimation of pitch and volume is unclear. Can you give more information?

(6) I advise to move this paragraph «  An electric condenser microphone ECM-674/9X from Sony Corporation was used as the system audio input device. For the output, a dialog robot was used with the built-in speaker of the humanoid  robot NAO v4 [37] used in the impression evaluation experiment of this paper. » to the section 4.2. Apparatus 

(7) The derivation procedure is unclear. Can you make a figure to explain that?

(8) You write "The effect size was estimated from the results, assuming an a level of 0.05 and a power (1 - b) of 0.80. ». Can you explain this sentence?

(9) You write "Scenarios were chosen rather  than chat because the reproducibility of the experiment can be maintained if the interaction use a predetermined scenario. » Can you explain this sentence?

(10) The section result must be analyse and comment.

(11)I highly recommend proofreading and correcting English.

Reviewer 3 Report

Interesting study, that provides insight in the role of vocal synchrony in robot acceptance/empathy recognition. The results are mainly presented via charts which could be more clarified in the text. Beside this the paper is well done, but focusses on a very specific aspect of HRI.

Author Response

Thank you for your comments.
We submit a revised manuscript.

Round 2

Reviewer 1 Report

This paper studied whether vocal synchronization in a robot affects emotional responses of those who interact with it. An experiment is presented with two conditions: vocal synchronization and de-synchronization. The robot was evaluated through a questionnaire, as well as using physiological signals gathered from the participants. The paper concludes that vocal synchronization can increase emphatic emotional responses.

The revised manuscript is significantly improved and many of my concerns are addressed. There are still a few points that are important to be addressed before acceptance of this paper:

(1) Research questions and hypotheses still need to be explained in a clear way. As it is commonly used, hypotheses can be listed, and later referred to in results and discussions. Currently, it is not clear if the responses to the questionnaire were posthoc and exploratory, or were hypothesized. The current section on research questions describes the terminology. That can be moved to another section, rather, research questions and hypotheses can be clarified there.

(2) Background still needs to be expanded. The authors have now included additional related articles, but they are all cited very briefly in the introduction. It is important to provide more details of the related work in the background, for example, to describe in more details what those articles did.

(3) The method for evaluating the accuracy of the constructed synchronization voice, which is explained in the response letter, needs to be added in the paper. Also, it should be clarified how and when the participants were recruited for the test, and why was it tested with a small sample and all male participants. The number seems to be low, so this should be also mentioned as a limitation.

(4) How were the participants asked for the qualitative comments?

(5) The paper needs to be further checked for grammar errors and typos (e.g.. “there were also significant difference between in the mean value between the..”)

(6) There are still some claims that are too strong and should be revised, for example: “We developed a robot that observes the features of human voice in real time and speaks on the basis…..” (because NAO robot is used in this study, and the proposed feature is not properly tested).

(7) The design of the experiment (between-participants design) needs to be added in the paper. Also, running a quick power analysis using g*power I get a larger sample required for the tests performed in this study with two conditions in a between-participants design. How were the parameters selected for getting 27 as the number of participants?

Reviewer 2 Report

Dear Authors,

Thank you for taking all my comments into consideration. Your paper has been very clearly improved, I congratulate you. I have 2 minors comments:

(1) I think that you can improve the quality of fig. 2.

(2) Please, can you give more details about the Fig. 3, for example, what do the colors correspond to?

(3) I give you 2 papers which can contribute to your discussion:

- Nadel, J., Simon, M., Canet, P., Soussignan, R., Blancard, P., Canamero, L., & Gaussier, P. (2006). Human responses to an expressive robot. In Procs of the Sixth International Workshop on Epigenetic Robotics. Lund University.

- Anzalone, S. M., Boucenna, S., Ivaldi, S., & Chetouani, M. (2015). Evaluating the engagement with social robots. International Journal of Social Robotics, 7(4), 465-478.

Best,
